# The chemokine receptor CCR8 is not a high-affinity receptor for the human chemokine CCL18

Khansa Hussain[1] , Herman D. Lim[2] , Shankar Raj Devkota[2], Barbara K. Kemp-Harper[3], J. Robert Lane[4,5], Meritxell Canals[4,5], James E. Pease[1‡]*, Martin J. Stone[2‡]*

1 National Heart and Lung Institute, Imperial College London, London, United Kingdom, 2 Monash Biomedicine Discovery Institute, and Department of Biochemistry and Molecular Biology, Monash University, Clayton, VIC, Australia, 3 Monash Biomedicine Discovery Institute, and Department of Pharmacology, Monash University, Clayton, VIC, Australia, 4 Division of Physiology, Pharmacology and Neuroscience, School of Life Sciences, Queen's Medical Centre, University of Nottingham, Nottingham, United Kingdom, 5 Centre of Membrane Protein and Receptors, Universities of Birmingham and Nottingham, Nottingham, United Kingdom

☯ These authors contributed equally to this work.
‡ JEP and MJS also contributed equally to this work.
* j.pease@imperial.ac.uk (JEP); martin.stone@monash.edu (MJS)

**Data Availability Statement:** All relevant data are within the manuscript.

**Funding:** This research was funded by a fellowship from the Government of Saudi Arabia and King

## Abstract

The primate-specific chemokine CCL18 is a potent chemoattractant for T cells and is expressed at elevated levels in several inflammatory diseases. However, the cognate receptor for CCL18 remains unconfirmed. Here, we describe attempts to validate a previous report that the chemokine receptor CCR8 is the human CCL18 receptor (Islam et al. *J Exp Med.* 2013, **210**:1889–98). Two mouse pre-B cell lines (4DE4 and L1.2) exogenously expressing CCR8 exhibited robust migration in response to the known CCR8 ligand CCL1 but not to CCL18. Similarly, CCL1 but not CCL18 induced internalization of CCR8 on 4DE4 cells. CCR8 expressed on Chinese hamster ovarian (CHO) cells mediated robust G protein activation, inhibition of cAMP synthesis and β-arrestin2 recruitment in response to CCL1 but not CCL18. Several N- and C-terminal variants of CCL18 also failed to stimulate CCR8 activation. On the other hand, and as previously reported, CCL18 inhibited CCL11-stimulated migration of 4DE4 cells expressing the receptor CCR3. These data suggest that CCR8, at least in the absence of unidentified cofactors, does not function as a high affinity receptor for CCL18.

## Introduction

The central hallmark of inflammation is the migration of leukocytes to injured or infected tissues. Leukocyte migration is regulated, in large part, by the interactions of chemokines (chemotactic cytokines), proteins secreted from the affected tissues, with chemokine receptors on the surfaces of leukocytes [1–3]. Chemokines are classified, based on the spacing of conserved cysteine residues, into two large subfamilies (CCL1-28 and CXCL1-17, in which 'L' indicates

Saud University (K.H.), National Health and Medical Research Council (NHMRC) Project Grant APP1140874 (M.J.S., M.C., J.R.L.) and NHMRC Ideas Grant 2012579 (M.J.S.). The funders had no role in study design, data collection and analysis, decision to publish, or preparation of the manuscript.

**Competing interests:** The authors have declared that no competing interests exist.

ligand) and two small subfamilies (CX$_3$CL1 and XCL1-2) (All chemokines and receptors referred to herein are the human forms unless indicated by the prefix 'm' for the murine forms.) [4]. Chemokine receptors are G protein-coupled receptors (GPCRs), classified, based on the classes of their cognate chemokine ligands (CCR1-10, CXCR1-6, CX$_3$CR1 and XCR1, in which 'R' indicates receptor) [4]. The spectrum of leukocytes recruited to particular tissues is dependent on the chemokines expressed in those tissues, the receptors differentially expressed on various types of leukocytes and the complex array of agonist (or antagonist) activity for each chemokine at each receptor. Therefore, to understand the biology of leukocyte recruitment in inflammation, it is important to identify cognate pairs of chemokines and chemokine receptors.

In this article we focus on CCL18, a chemokine that is unique to primates, with no apparent orthologue in rodents, but whose cognate receptor(s) has/have remained unconfirmed. Prior to the introduction of the systematic naming convention, CCL18 was identified independently by four groups, who each assigned it a different acronym. Hieshima and coworkers found CCL18 to be constitutively expressed at high levels in the lung and named it pulmonary and activation-regulated chemokine (PARC) [5]. Adema et al. found that CCL18 cloned from a human dendritic cell cDNA library could recruit naive T cells, and designated it dendritic cell CC chemokine (DC-CK1) [6]. Separately, two other groups identified CCL18 from cDNA libraries and named it macrophage inflammatory protein-4 (MIP- 4) or alternative macrophage activation-associated CC chemokine-1 (AMAC-1) due to its expression by macrophages and sequence similarity to other MIPs [7, 8]. Subsequently, CCL18 has been reported to be expressed by alternatively activated (M2) macrophages [8] and also by CXCL4-activated macrophages [9], although we and others were unable to replicate this finding [10, 11]. CCL18 is constitutively expressed in healthy human plasma at relatively high levels (10–72 ng/mL) [12]. However, elevated levels of CCL18 have been reported in several inflammatory diseases, including atherosclerosis [13], pulmonary disorders [14] dermatitis [15] and HIV infection [16]. Most notably, serum CCL18 levels were dramatically elevated (an average of 29-fold higher than controls) in patients with Gaucher's disease, a lysosomal storage disorder, leading to its rapid acceptance as a biomarker for diagnosis [17].

A variety of independent studies have verified that CCL18 can promote the chemotaxis of various T cell populations, including naïve T cells, T helper 2 (Th2) cells, regulatory T cells (Tregs), and both CD4$^+$ and CD8$^+$ T lymphocytes, as well as NK cells, B lymphocytes and basophils [5, 6, 18–24]. However, efforts to identify the target receptor(s) for CCL18 have met with limited success. CCR6 was reported as a human CCL18 receptor in a conference abstract [25]. Later, Chen et al. reported that CCL18 is highly expressed by tumour-associated macrophages and promoted the invasiveness of breast cancer cells, an effect mediated by membrane-associated phosphatidylinositol transfer protein 3, PITPNM3 [26]. However, subsequent studies have failed to detect PITPNM3 on leukocytes, suggesting they are not responsible for the effects of CCL18 on these cells [24]. Moreover, we previously found that neither CCR6 nor PITPNM3 functioned as a CCL18 receptor in a mammalian cell transfection system [27]. Catusse et al. reported that, in pre-B acute lymphocytic leukemia cells, the receptor GPR30 mediated the ability of CCL18 to suppress Ca$^{2+}$ mobilization and chemotaxis responses to CXCL12, but this does not account for the induction of such responses in T cells by CCL18 [28]. In an apparent breakthrough, Islam et al. reported that CCL18 activated human CCR8 exogenously expressed in a mouse pre-B cell line (4DE4 cells) and that CCL18-stimulated chemotaxis of Th2 cells was dependent on CCR8 expression and blocked by an anti-CCR8 antibody [23]. Moreover, the same group found that mCCL8 activated mCCR8 and therefore designated mCCL8 as a murine functional analogue of human CCL18 [23, 29]. However, Barsheshet reported that CCL18 failed to induce calcium flux in CCR8-transfected CHO cells

[30]. Moreover, to our knowledge there have been no further reports validating CCR8 as a CCL18 receptor.

In light of the significant roles of CCL18 in leukocyte biology and inflammatory diseases, we have undertaken systematic analysis of whether CCR8 functions as a valid CCL18 receptor. The data presented herein represent independent studies and complementary approaches from two laboratories. In one approach, we found that the previously-validated CCR8 agonist CCL1, was able to stimulate chemotaxis and endocytosis via CCR8 expressed in mouse pre-B cells but that CCL18 did not elicit such responses. In an alternative approach, we found that Chinese hamster ovary (CHO) cells transfected to express CCR8 were able to support both G protein-mediated signalling and β-arrestin2 recruitment in response to CCL1 but not CCL18. Similarly, mCCR8 did not respond to mCCL8. Together, these data suggest that CCR8 is not sufficient to act as a cognate receptor for CCL18.

## Materials and methods

### Materials

CCL18(2–69) (cat. no. 300–34), CCL1 (cat. no. 300–37), and mCCL8 (cat. no. 250–14) were purchased from PeproTech (Cranbury, NJ;). CCL18(1–69) (cat. no. 394-PA) and mCCL1 (cat. no. 845-TC) were purchased from R&D Systems (Minneapolis, MN, USA;). AF687-labelled CCL18 (AF-CCL18) was purchased from Almac (Craigavon, UK).

The anti-CCR8 primary antibody 433H was a kind gift of Professor David Cousins (University of Leicester). The IgG$_{2a}$ isotype control UPC-10 was obtained from Sigma-Aldrich. The goat anti-mouse FITC labeled F(ab')2 secondary antibody was purchased from Invitrogen.

Dulbecco's modified Eagle's medium (DMEM), RPMI 1640 medium, fetal bovine serum (FBS), 100x MEM non-essential amino acid solution, hygromycin B, and D-luciferin were sourced from Life Technologies (Carlsbad, CA, USA). Penicillin and streptomycin were obtained from Life Technologies or from Sigma- Aldrich (St. Louis, MO, USA). Coelenterazine h was obtained from Prolume (Pinetop, AZ). Cloning enzymes and buffers were obtained from New England Biolabs (Ipswich, MA, USA). Other reagents were sourced from Sigma-Aldrich (St. Louis, MO) or ThermoFisher Scientific (Waltham, MA, USA). White 96-well culture plates were purchased from Revvity (Waltham, MA, USA).

### Cell lines and cell culture

The parental 4DE4 cell line was a kind gift form Dr Louis Staudt, (NIH, Bethesda, MD, USA). Two 4DE4 clonal lines stably expressing human CCR8 (4DE4-CCR8) and human CCR3 (4DE4-CCR3) were generated in the laboratory of Dr Philip Murphy, (NIH, Bethesda, MD, USA) and have been described previously [31, 32]. The parental L1.2 cell line has been previously described [33].

L1.2 and 4DE4 cells were maintained, at 37°C and 5% CO$_2$, at 0.5–1.0 x 10$^6$ cells/mL in RPMI 1640 medium with (Glutamax-I, 25mM HEPES) supplemented with 10% heat-inactivated FBS, 100 U/mL penicillin/streptomycin, 1x non-essential amino acids, 1 mM sodium pyruvate and 0.001 mM β-mercaptoethanol (referred to as complete media [34]. Transfected L1.2 and 4DE4 cells stably expressing CCR8 (L1.2-CCR8 and 4DE4-CCR8) were cultured in complete media supplemented with G418 (1 mg/mL) to maintain selection.

L1.2 cells were transiently transfected with plasmid encoding CCR8 as previously described [34]. Transfected cells were then incubated for 3–5 hour (37°C, 5% CO$_2$), sodium butyrate was added to a final concentration of 10 mM, and the cells were cultured for 18 to 24 hours before evaluation of receptor expression by flow cytometry and use in functional assays.

For expression in CHO cells, the gene for human CCR8 was obtained from the cDNA Resource Center (www.cDNA.org) and the gene for mouse CCR8 (mCCR8) was a gift from Remy Robert (Monash University, Australia). These genes were subcloned into a modified pcDNA5/FRT/TO vector and used to stably transfect FlpInCHO cells as described [35]. The parental FlpInCHO cells were maintained in DMEM supplemented with 5% FBS, 100 U/mL penicillin and streptomycin. The FlpInCHO-CCR8 and FlpInCHO-mCCR8 cell lines were selected with 200 μg/mL hygromycin B and used for G protein activation and cAMP inhibition assays.

## Expression, purification and analysis of CCL18 variants

The gene encoding CCL18 was synthesized as a gBlock by Integrated DNA Technologies, and subcloned into an expressed vector derived from pET-28a (Novagen), which encoded an N-terminal His$_6$ tag followed by a Small Ubiquitin-like Modifier (SUMO) tag. Recombinant expression vectors were verified by Sanger sequencing then used to transform Rosetta-gami 2 (DE3) *E. coli* competent cells. Transformants were cultured in 2YT medium containing 30 μg/mL kanamycin at 37˚C until the optical density at 600 nm reached 0.6–0.7. Expression was induced by the addition of isopropyl β-D-1-thiogalactopyranoside to a final concentration 0.5 mM, and culture was continued overnight at 20˚C. After harvesting by centrifugation (5,000 g, 10 min, 4˚C), cells were resuspended in 50 mL of lysis buffer containing 20 mM Tris pH 8.0, 500 mM NaCl, 10% glycerol, lysed by sonication, and the lysate was clarified by centrifugation (29,000 g, 40 min, 4˚C). The His$_6$-SUMO-CCL18 fusion protein was initially purified by loading in lysis buffer onto a 5 mL His-Trap FF nickel column attached to an ÄKTA pure chromatography system (Cytiva) and elution with the lysis buffer containing 500 mM imidazole. Eluted fractions containing the fusion protein were combined and dialysed overnight in lysis buffer then treated with ULP1 protease (prepared in-house, 60 min, 25˚C) to cleave the desired CCL18 variant from the His$_6$-SUMO tag, which were separated from each other by passage through the His-Trap FF nickel column. The non-binding fractions (containing CCL18) were combined and concentrated using Amicon® 3kDa centrifugal filters, and further purified by size exclusion chromatography in 10 mM HEPES pH 7.5, 150 mM NaCl, 5% glycerol buffer using a Superdex 75 size column attached to an ÄKTA pure chromatography system (Cytiva). Fractions containing protein of the expected size were combined, concentrated and analysed by analytical reversed-phase high-performance liquid chromatography on an Agilent Technologies 1200 series instrument equipped with a 214TP C4 (250 x 10 mm, 5 μm) column (Vydac), pre-equilibrated with 0.001% trifluoracetic acid in water and eluted using a 1% per min gradient of acetonitrile containing 0.001% trifluoracetic acid. Protein molecular masses were confirmed by electrospray ionisation mass spectrometry performed at the Monash Proteomics and Metabolomics Platform.

## Cell surface receptor expression by flow cytometry

Flow cytometry buffer was prepared by supplementing a 500 mL bottle of 1X Dulbecco's phosphate buffered saline with 1.25 g of bovine serum albumin and 500 μl of a 10% sodium azide solution and stored at 4˚C until needed. Cells (~0.2–0.5 x 10$^6$) were collected by centrifugation (300 g, 5 min, room temperature). The supernatant was removed and the cells were gently resuspended in staining buffer (100 μL) containing the anti-CCR8 antibody 433H or the isotype control (10 μg/mL). After incubation on ice (up to 15 min), cells were washed with staining buffer (500 μL), resuspended in staining buffer (100 μL) containing goat anti-mouse FITC labeled F(ab')2 secondary antibody (1:20 dilution), and incubated on ice for 15–30 min. Cells were washed again with staining buffer (500 μL) and resuspended in staining buffer (500 μL). For dead cell exclusion, TOPRO-3 (Invitrogen) was added to the staining buffer at a dilution

of 1:10000. Cells were transferred to flow cytometry tubes and analysed on a FACSCalibur Flow Cytometer (Becton Dickinson, Oxford, UK) using CellQuest Pro software following the manufacturer's instruction. 10,000 events were typically acquired. Data are presented as a percenatge of specific CCR8 expression in the absence of treatment.

### Chemotaxis assays

Chemotaxis assays were performed using 96-well ChemoTx System chemotaxis plates (101–5, pore size 5 μm) obtained from Neuroprobe Inc. (Rockville, MD, USA). The protocol was as previously described [34] with assays incubated in a humidified tissue-culture incubator for 5 h at 37˚C, 5% $CO_2$. At the conclusion of the incubation, cells on the top of the membrane were removed by gentle scraping and cells migrating into the lower well were pelleted in the wells by centrifugation (300 g, 5 min, room temperature). The cells were then transferred to an opaque white plate, via a filter funnel (Neuroprobe), by centrifugation (300 g, 5 min, room temperature). To each well, 30 μL of the live cell dye CellTiter Glo® (Promega, Southampton, UK) was added and luminescence counts read using a TopCount NXT (Perkin Elmer). The chemotactic index was calculated as the average cell migration calculated at each chemokine concentration divided by the average cell migration observed to buffer alone.

### Receptor internalization assays

4DE4-CCR8 cells were resuspended in ice-cold RPMI medium at a concentration of $5 \times 10^6$ cells/mL. Duplicate tubes containing 25 μL of buffer (negative control) or the indicated concentrations of chemokines were placed on ice and then 25 μL of cell suspension (250,000 cells) was added with gentle mixing. Tubes were either left on ice or incubated at 37˚C for 30 min. Then processed for flow cytometry using the 433H CCR8 mAb or the isotype control as described earlier. CCR8 cell surface expression was normalized to the CCR8 levels on untreated cells, following subtraction of non-specific binding of the isotype control.

### AF-CCL18 binding assays

Cells (were collected by centrifugation (300 g, 5 min, room temperature)and resuspended in flow cytometry buffer at $20 \times 10^6$ /mL. Tubes containing 25μL of varying concentrations of AF-CCL18 were placed on ice and 25μL of the cell suspension added with gentle mixing. Following a 30 minute incubation on ice, cells were then washed with 500 μL of staining buffer, pelleted by centrifugation and resuspended in fresh staining buffer prior to being analyzed on the flow cytometer. Mean fluorescence data from 5 independent experiments are reported.

### G protein activation assays

FlpInCHO-CCR8 or FlpInCHO-mCCR8 cells were transiently transfected to express biosensors for G-protein activation, as described [35]. One day after transfection, the cells were transferred into a white 96-well plate (50,000 cells/well in 100 μL medium) and the plate was incubated overnight. The cells were then stimulated, in a total assay volume of 80 μL, with the indicated concentrations of chemokines in Hank's balanced salt solution (HBSS). The BRET signals were measured 10 min after stimulation, using a PHERAstar microplate reader (BMG Labtech, Ortenberg, Germany) to detect fluorescence at 450–500 nm and 515–560 nm.

### cAMP inhibition assays

FlpInCHO-CCR8 or FlpInCHO-mCCR8 cells were transiently transfected to express the CAMYEL biosensor [36]. One day after transfection, the cells were transferred into a 96-well

plate then treated with chemokines in the presence of forkolin and BRET signals were detected as described above for G protein activation assays [35].

### β-arrestin recruitment assays

The DNAs encoding CCR8 and mCCR8 tagged at their N-termini with a C-myc epitope tag and at their C-termini with Renilla luciferase variant 8 (RLuc8) were constructed as described for other chemokine receptors tagged in the same manner [35]. The DNA encoding β-arrestin2, N-terminally tagged with yellow fluorescent protein (YFP) [37], was a gift from Prof. Kevin Pfleger (University of Western Australia). These two DNA constructs were transiently co-expressed in parental FlpInCHO cells, as described [35]. One day after transfection, the cells were transferred (50,000 cells/well in 100 μL medium) into a 96-well plate, incubated overnight, then stimulated in a total assay volume of 80 μL, with the indicated concentrations of chemokines and 5 μM coelenterazine-h in HBSS. The BRET signals were detected, as described above for G protein activation assays.

## Results

### CCL18 does not stimulate CCR8-mediated chemotaxis

Leukocytes expressing chemokine receptors typically migrate in response to cognate ligands of that receptor, exhibiting a bell-shaped dependence on chemokine concentration. Using a 4DE4 (mouse pre-B cell) cell line stably-transfected with CCR8, which had originated from one of our laboratories [31], Islam et al. reported a typical bell-shaped concentration response to CCL18, with peak response at ~10 nM [23]. In an attempt to replicate this observation, we evaluated both the expression of CCR8 and the chemotactic response of this cell line to CCL18 and to CCL1, the only other human chemokine known to be an agonist of CCR8. Using flow cytometry, we verified robust expression of CCR8 in the transfected 4DE4 cell line, as expected (Fig 1A). In a chemotaxis assay, the 4DE4-CCR8 transfectants exhibited a robust bell-shaped response to CCL1, with a peak at ~10 nM (Fig 1B). Moreover, CCL18 also stimulated a strong chemotactic response, albeit at substantially higher concentrations (Fig 1B). However, a weaker response to CCL18 was also observed in the parental 4DE4 cell line (Fig 1B), raising the possibility that the CCL18 response was not mediated by CCR8 but instead by an endogenous, low affinity receptor. Consistent with this latter possibility, incubation with the virally-encoded CCR8 inhibitor MC148 completely blocked 4DE4-CCR8 chemotaxis in response to 1 nM CCL1 but had no effect on the response of parental 4DE4 cells to 2 μM CCL18 (Fig 1C).

Notably, Islam et al. reported that the CCR8-transfected 4DE4 cell line responded better to CCL18 when grown in relatively low concentrations (0.1–0.4 mg/ml) of Geneticin (G418) [23]. Therefore, we further evaluated chemotaxis of this cell line after adaptation of two separate clones to 0.1 mg/mL G418. For both clones, the robust response to CCL1 was retained but there was no significant response to CCL18 (Fig 2A and 2B). To further assess the ability of CCR8 to function as a CCL18 receptor, we utilised another murine pre-B cell line, L1.2 [33, 38], either stably or transiently transfected to express CCR8 [39]. In contrast to the strong chemotactic responses to CCL1 in both transfectants, CCL18 had no significant chemotactic activity at concentrations as high as 0.5 μM (Fig 2C and 2D). Taken together, our results indicate that CCR8 does not mediate chemotaxis in response to CCL18 in murine pre-B cells.

### CCL18 does not stimulate CCR8 internalisation

Following incubation with their cognate ligands, chemokine receptors typically undergo endocytosis [40]. To further clarify whether CCL18 is a *bona fide* ligand of CCR8, we used flow

**A**

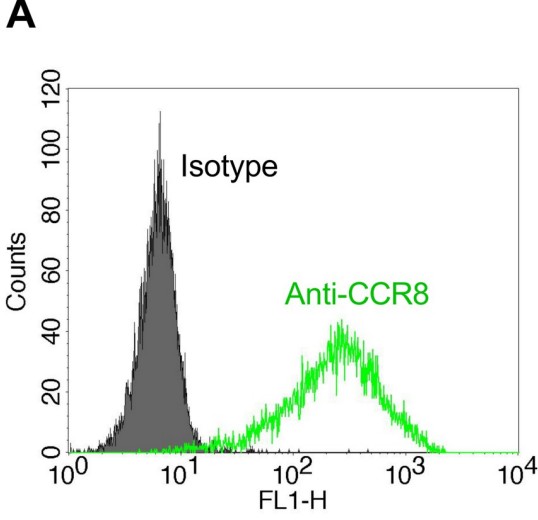

**B**

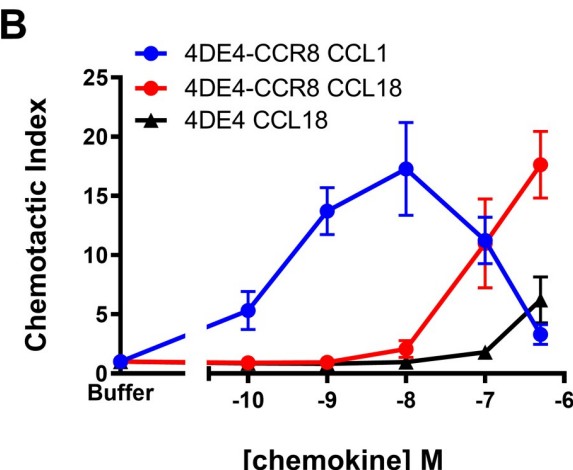

**C**

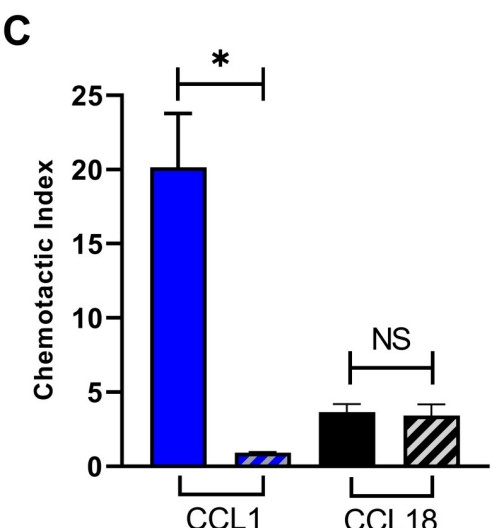

**Fig 1. CCL18 is a poorly potent recruiter of 4DE4-CCR8 cells and the parental 4DE4 line in chemotaxis assays.**
(A) Cell surface levels of CCR8 expressed stably in the 4DE4 cell line maintained in 0.1 mg/mL G418. The green line denotes detection with an anti-CCR8 mAb and the solid histogram reflects isotype control staining. (B) Comparative chemotaxis responses of the same 4DE4-CCR8 line to CCL1 (blue) and CCL18 (red) and the parental 4DE4 line (maintained without G418) to CCL18 (black). (C) Inhibition of chemotactic responses to 1 nM CCL1 and 2 μM CCL18 from 4DE4-CCR8 cells (blue) and the parental 4DE4 line (black), respectively. Striped bars denote pre-incubation of cells for 30 min at 37˚C with 500 nM MC148. Data represent mean chemotactic indices ± S.E.M. from 3 independent experiments. * denotes $p < 0.05$ as examined by a paired t-test.

cytometry of the 4DE4-CCR8 cell line to monitor the internalisation of CCR8 in response to chemokines. Cell surface CCR8 was monitored using the 433H monoclonal antibody, or iso-type control, following incubation with chemokine for 30 minutes. As anticipated from previous work [39], 100 nM CCL1 treatment resulted in significant (~75%) apparent internalisation of CCR8, as indicated by the loss of 433H staining, whereas 100 nM CCL17 (negative control) gave no significant internalisation (Fig 3A and 3B). Treatment with 1 μM CCL18 resulted in no significant internalisation (Fig 3B), again indicating that CCL18 is not an agonist of CCR8. When the binding of 433H to 4DE4-CCR8 in the presence of 100 nM CCL1 was compared at

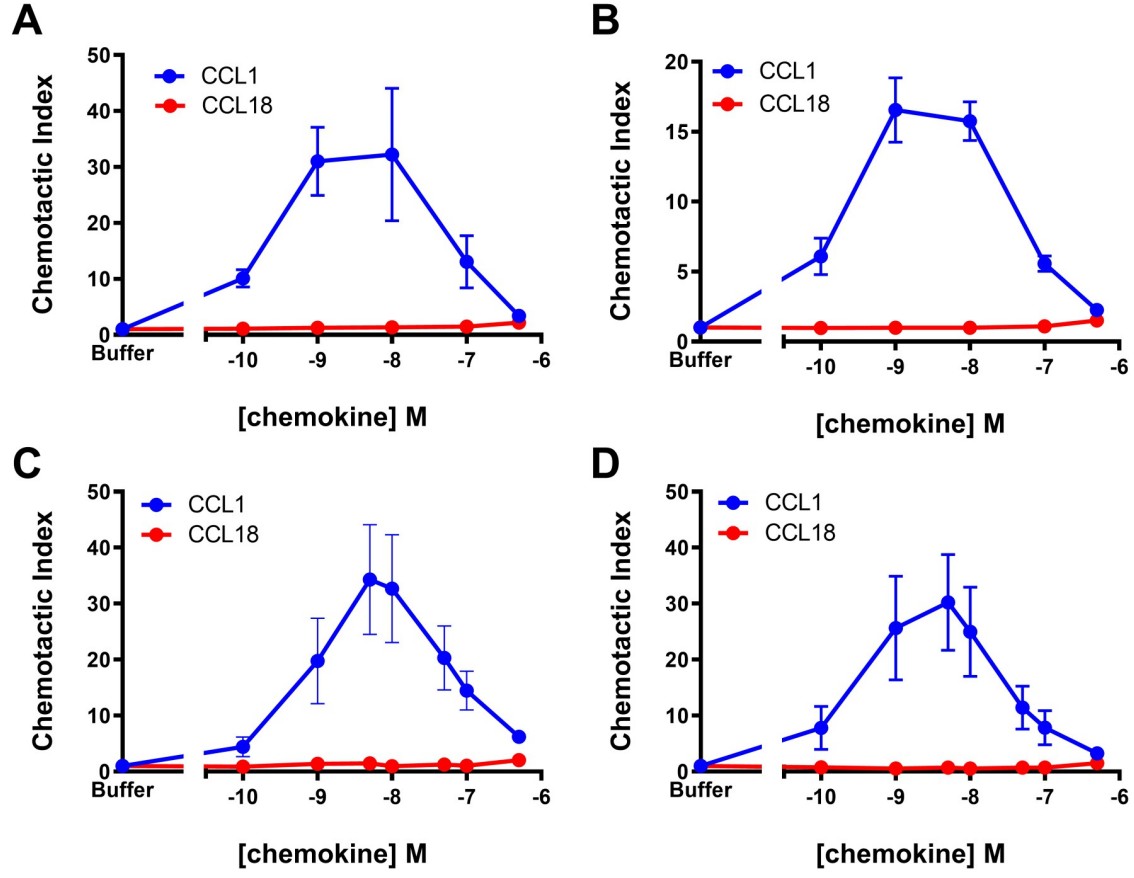

**Fig 2. CCL18 is a poorly potent recruiter of various mouse pre-B cells expressing CCR8.** (A, B) Chemotactic responses to CCL1 (blue) and CCL18 (red) from two independently rederived clones of the 4DE4-CCR8 cell line maintained in 0.1 mg/mL G418. (C, D) Chemotactic responses to CCL1 (blue) and CCL18 (red) from L1.2 cells stably (C) or transiently (D) expressing CCR8. Data represent mean chemotactic indices ± S.E.M. from 3 (panels A -B) or 4 (panels C-D) independent experiments.

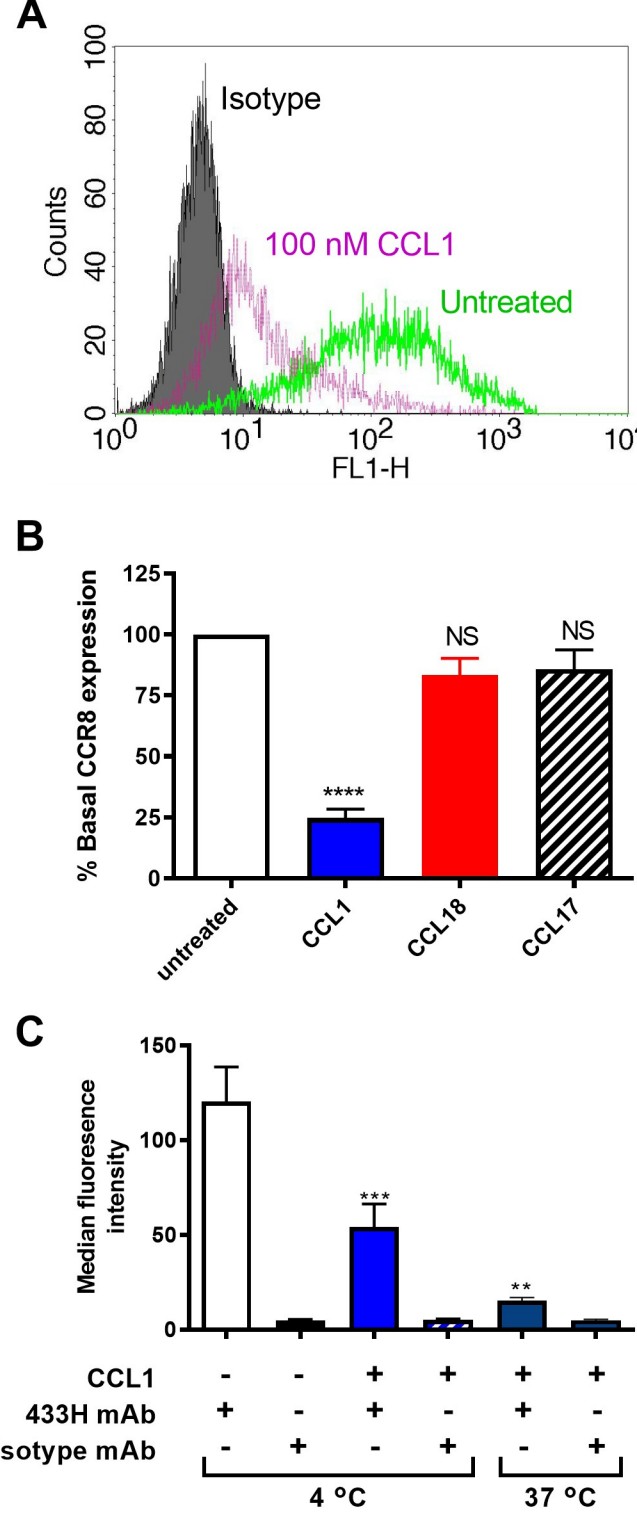

**Fig 3. CCL18 fails to induce significant endocytosis of CCR8 in transfectants.** (A) Representative cell surface levels of CCR8 on the 4DE4-CCR8 stable cell line which were untreated (green line) or incubated with 100 nM CCL1 (magenta line) for 30 min at 37˚C, then assayed by flow cytometry. Isotype staining of untreated cells is shown as a

comparator (filled histogram). (B) Levels of CCR8 expression on the 4DE4-CCR8 cells following incubation with CCL1 (100 nM), CCL18 (1 µM), or CCL17 (100 nM) for 30 minutes at 37°C. Data represent mean % basal CCR8 expression levels ± S.E.M. from 6 independent experiments. **** represents p<0.0001 and NS represents no significant difference compared with untreated cells, as examined by one-way ANOVA and Bonferroni's multiple comparisons test. (C) CCR8 expression detected by 433H (anti-CCR8) or isotype control on 4DE4-CCR8 cells following incubation with either buffer for 30 min at 4°C or 100 nM CCL1, for 30 min at 4°C or 37°C. Data show median fluorescence levels ± S.E.M. from 6 independent experiments. *** represents p<0.001 and ** represents p<0.01 compared with buffer treated cells stained with 433H, as examined by one-way ANOVA and Bonferroni's multiple comparisons test.

4°C and 37°C (Fig 3C), it was evident that incubation with CCL1 at 4°C lead to a significant loss of the CCR8 signal suggesting that CCL1 binding to CCR8 partially masks 433H detection. However, a further loss of signal was observed with CCL1 incubation at 37°C, suggesting that CCL1 also promotes CCR8 endocytosis. Taken together, our data suggest that CCL1 but not CCL18 can productively interact with CCR8.

## CCL18 competitively inhibits CCR3 but not CCR8

Nibbs and co-workers have reported that CCL18 is an antagonist of the chemokine receptor CCR3 [41]. Therefore, considering the lack of activity observed for CCL18 at CCR8, we tested the ability of CCL18 to inhibit the activity of CCR3, using 4DE4 cells stably transfected with CCR3 [32]. Flow cytometry confirmed expression of CCR3 in these cells (Fig 4A). Moreover, 1 nM CCL11, a cognate agonist of CCR3, stimulated a robust chemotactic response, which was inhibited in a concentration-dependent manner by CCL18, with a 50% inhibitory concentration below 50 nM (Fig 4B). In contrast, an identical concentration range of CCL18 had no significant effect on the migration of CCR8 transfectants towards 1 nM CCL1 (Fig 4C).

To more directly evaluate binding of CCL18 to both CCR3 and CCR8, we performed flow cytometry of stably transfected 4DE4 cells (or untransfected controls) using CCL18 tagged at the C-terminus with Alexa Fluor 647 (AF-CCL18). AF-CCL18 bound significantly and in a concentration-dependent manner to 4DE4-CCR3 transfectants but exhibited no significant binding to 4DE4-CCR8 transfectants in comparison to the parental 4DE4 cell line (Fig 4D). These data confirm that CCL18 is a potent antagonist of CCR3 but, at best, a low affinity ligand for CCR8.

## CCL18 does not stimulate CCR8-mediated signalling in CHO cells

To eliminate the possibility of interference from endogenous receptors, chemokine receptor activation and signalling is often monitored using common mammalian cell lines that do not endogenously express chemokine receptors. Thus, in a complementary approach to the above experiments, we investigated the ability of CCR8 to signal in response to CCL18 in a FlpIn CHO cell line stably transfected to express CCR8. Signalling in response to CCL18(2–69) (see below) or CCL1, a known agonist of CCR8, was then monitored using three measures of chemokine receptor signalling–G protein activation, inhibition of cyclic AMP synthesis (cAMP inhibition) and β-arrestin2 recruitment.

Upon stimulation by their cognate ligands, GPCRs typically undergo conformational changes resulting in dissociation of their bound heterotrimeric G proteins into Gα and Gβγ subunits. The Gβγ subunit may then bind to G protein receptor kinases (GRKs). Thus, we measured G protein activation by monitoring the association of labelled Gβγ with a labelled GRK fragment in a bioluminescence resonance energy transfer (BRET) assay [35]. As expected, CCL1 robustly stimulated G protein activation (monitored using isoform Gαi2) in FlpIn CHO cells expressing CCR8, with a half-maximal effective concentration ($EC_{50}$) of 46

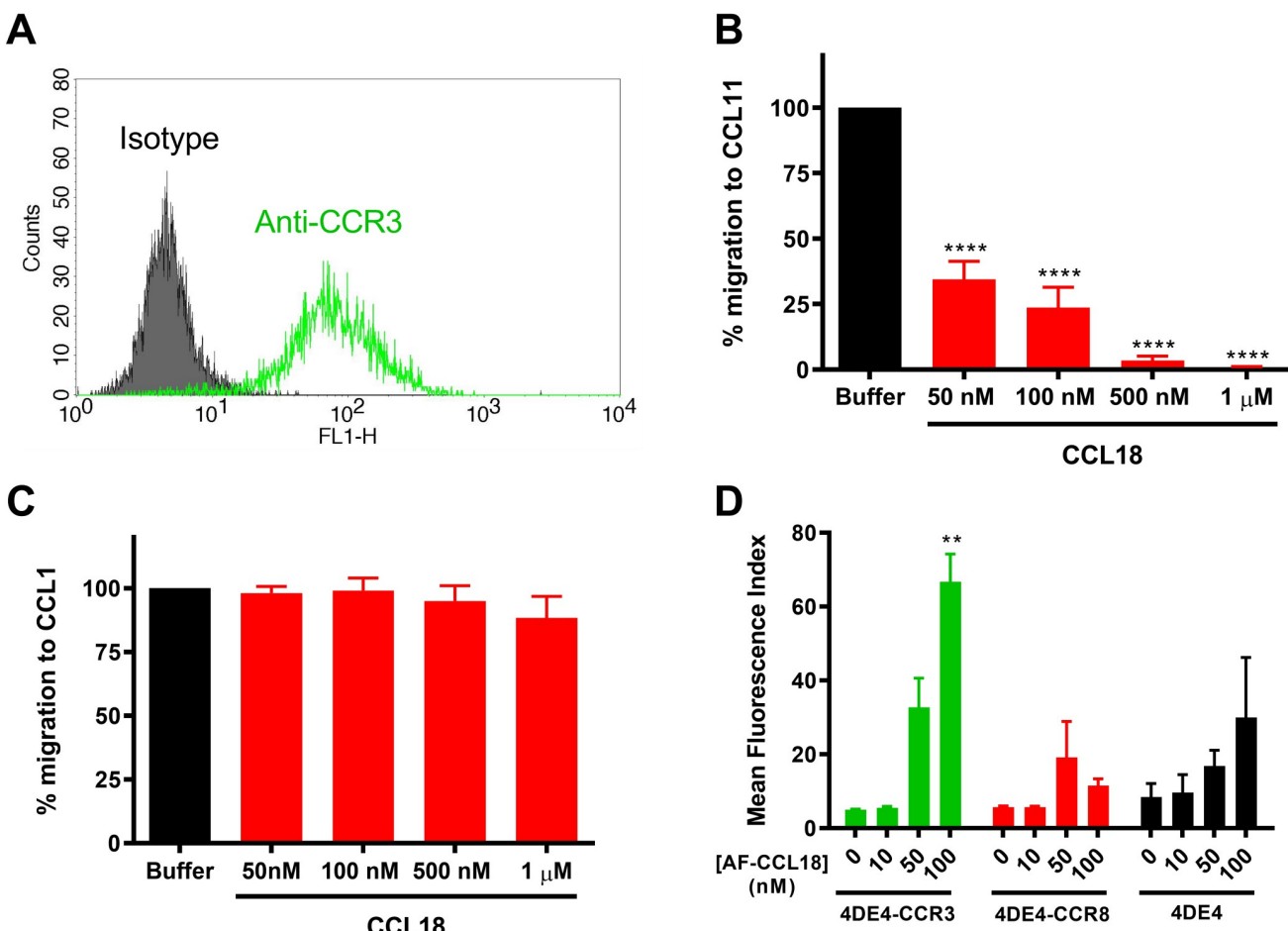

**Fig 4. CCL18 can antagonize chemotaxis of 4DE4-CCR3 but not 4DE4-CCR8 transfectants.** (A) Representative cell surface levels of CCR3 (green line) on the 4DE4-CCR3 stable cell line. Isotype control staining is shown as a shaded histogram. (B, C) Inhibition of chemotactic responses of 4DE4-CCR3 (B) and 4DE4-CCR8 (C) transfectants to a fixed 1 nM concentration of CCL11 or CCL1, respectively, by increasing concentrations of CCL18. Data shown are mean % of migrating levels ± S.E.M. from 5 independent experiments. **** represents p<0.0001 when compared with responses to buffer. (D) Binding of increasing concentrations of AF-CCL18 to 4DE4-CCR3 (green), 4DE4-CCR8 (red) and parental 4DE4 (black) cells. Data represent mean fluorescent indices ± S.E.M. from 5 independent experiments. ** represents p<0.01 when compared with buffer treatment, as examined by one-way ANOVA and Bonferroni's multiple comparisons test.

nM ($pEC_{50}$ (M) = 7.34 ± 0.12; Fig 5A). However, CCL18 failed to stimulate any significant G protein activation in these cells at concentrations up to 300 nM (Fig 5A).

Chemokine receptors typically couple to inhibitory G proteins (Gi) such that receptor activation gives rise to release of Gαi and inhibition of cAMP synthesis catalysed by adenylyl cyclase. Therefore, we used a cAMP BRET-based biosensor (CAMYEL) to monitor the inhibition of forskolin (Fsk)-stimulated cAMP synthesis after stimulation of CCR8-expessing FlpIn CHO cells with chemokines [35]. As observed in the G protein activation assay, CCL1 robustly inhibited cAMP synthesis ($EC_{50}$ = 26 nM; $pEC_{50}$ (M) = 7.59 ± 0.10), whereas CCL18 had no effect up to a concentration of 1 μM (Fig 5B).

In addition to initiating G protein-mediated pathways, activation of GPCRs, including chemokine receptors, often induces the recruitment of β-arrestins. Moreover, atypical chemokine receptors induce β-arrestin recruitment in the absence of detectable G protein-mediated

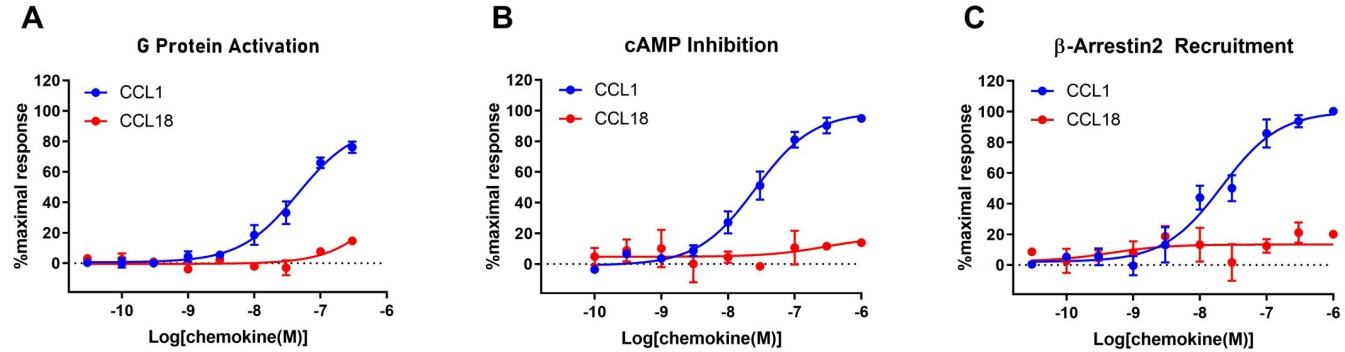

**Fig 5. CCR8 mediates signal transduction in response to CCL1 but not CCL18.** Concentration-response curves for chemokine-induced (A) G protein activation, (B) cAMP inhibition and (C) β-arrestin2 recruitment in Flp-In CHO cells stably expressing CCR8. The presented data are mean ± S.E.M. of at least three independent experiments, each performed in triplicate.

signalling. To determine whether, CCL18 stimulates β-arrestin recruitment to CCR8, we used a BRET assay in which β-arrestin2 and CCR8 were tagged with a BRET donor and acceptor, respectively [35]. Once again, CCL1 yielded a robust, concentration-dependent response ($EC_{50}$ = 21 nM; $pEC_{50}$ (M) = 7.67 ± 0.15), whereas CCL18 had no effect up to 1 μM (Fig 5C).

In summary, the above three assays consistently indicate that CCL1 is an agonist of CCR8 whereas CCL18 is not.

## Alternative forms of CCL18 do not activate CCR8

Chemokines are naturally expressed with a signal sequence that is removed proteolytically during secretion. Moreover, many secreted chemokines are subsequently trimmed at the N-terminus through the action of extracellular proteases, and these modifications can influence the ability of the chemokine to activate chemokine receptors, potentially increasing or decreasing potency and/or efficacy [42–44]. These factors create uncertainties regarding the biologically relevant N-terminal sequence(s) of chemokines.

CCL18 has been isolated from human plasma and from the conditioned media of stimulated peripheral blood mononuclear cells in several different forms, including: CCL18(1–69) (the "full-length" sequence); CCL18(1–68); CCL18(3–69); and CCL18(4–69) [45, 46] (Fig 6A). Creating more confusion, CCL18 is available commercially in different forms: CCL18(2–69) from PeproTech and CCL18(1–69) from R&D Systems (Fig 6A). Islam et al. reported using CCL18 from R&D Systems [23] and the experiments in Figs 1–4 were performed using CCL18 (1–69) from R&D Systems, whereas those in Fig 5 were performed using CCL18(2–69) from PeproTech.

To investigate whether differences in N-terminal sequences account for the differing reports of CCL18 activity at CCR8, we expressed in *E. coli* and purified four different forms of CCL18: CCL18(1–69), CCL18(2–69), CCL18(3–69) and CCL18(1–68); we also attempted to produce CCL18(4–69) but the resulting sample was insufficiently pure so omitted from subsequent experiments. The purity and identity of the CCL18 variants were verified by reversed-phase HPLC and mass spectrometry, respectively (Fig 6B and 6C). Nevertheless, as observed for commercial CCL18(2–69) (Fig 5A), none of these forms of CCL18 (at 100 nM) significantly stimulated G protein activation in CCR8-expressing FlpIn CHO cells, whereas the positive control (100 nM CCL1) stimulated a significant response (Fig 6D). Furthermore, none of

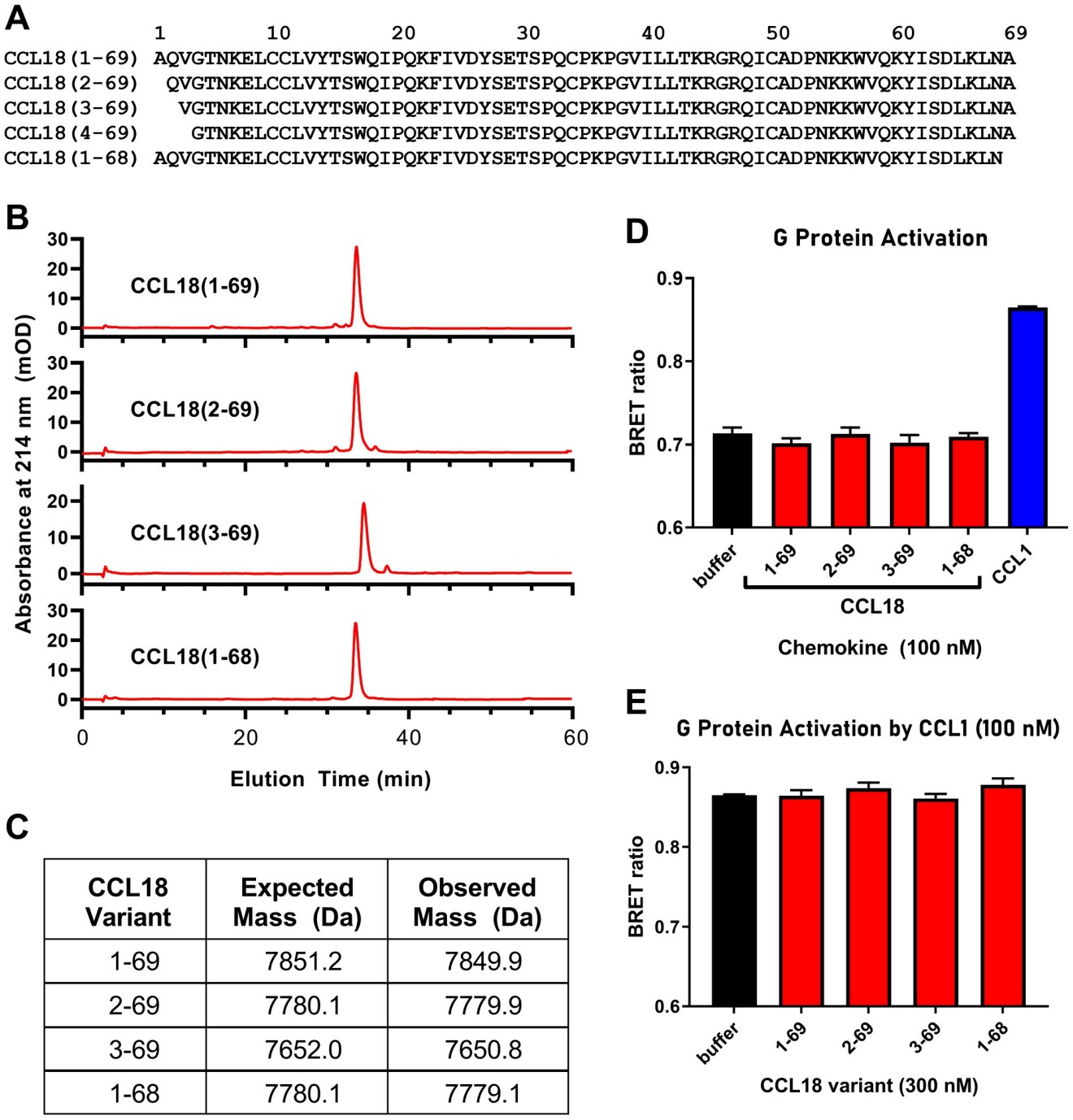

**A**

```
                1        10        20        30        40        50        60      69
CCL18(1-69)  AQVGTNKELCCLVYTSWQIPQKFIVDYSETSPQCPKPGVILLTKRGRQICADPNKKWVQKYISDLKLNA
CCL18(2-69)   QVGTNKELCCLVYTSWQIPQKFIVDYSETSPQCPKPGVILLTKRGRQICADPNKKWVQKYISDLKLNA
CCL18(3-69)    VGTNKELCCLVYTSWQIPQKFIVDYSETSPQCPKPGVILLTKRGRQICADPNKKWVQKYISDLKLNA
CCL18(4-69)     GTNKELCCLVYTSWQIPQKFIVDYSETSPQCPKPGVILLTKRGRQICADPNKKWVQKYISDLKLNA
CCL18(1-68)  AQVGTNKELCCLVYTSWQIPQKFIVDYSETSPQCPKPGVILLTKRGRQICADPNKKWVQKYISDLKLN
```

**B** C4 reversed-phase HPLC traces for purified CCL18 variants, Absorbance at 214 nm (mOD) vs Elution Time (min), for CCL18(1-69), CCL18(2-69), CCL18(3-69), CCL18(1-68).

**C**

| CCL18 Variant | Expected Mass (Da) | Observed Mass (Da) |
|---|---|---|
| 1-69 | 7851.2 | 7849.9 |
| 2-69 | 7780.1 | 7779.9 |
| 3-69 | 7652.0 | 7650.8 |
| 1-68 | 7780.1 | 7779.1 |

**D** G Protein Activation. BRET ratio for buffer, CCL18 (1-69, 2-69, 3-69, 1-68), CCL1. Chemokine (100 nM).

**E** G Protein Activation by CCL1 (100 nM). BRET ratio for buffer, CCL18 variant (1-69, 2-69, 3-69, 1-68). CCL18 variant (300 nM).

**Fig 6. Four CCL18 variants fail to activate or inhibit G protein activation mediated by CCR8.** (A) Sequences of CCL18 variants. (B) C4 reversed-phase HPLC traces for purified CCL18 variants. (C) Mass spectrometry data for the purified CCL18 variants. Expected masses assume two disulfide bonds. Observed masses were obtained from the 5+ charged peaks in electrospray ionization spectra. (D) G protein activation data for CCL18 variants (red) and CCL1 (positive control, blue), each at 100 nM. (E) The CCL18 variants (red), each at 300 nM, fail to inhibit G protein activation stimulated by 100 nM CCL1. The data in panels D and E were obtained using Flp-In CHO cells stably expressing CCR8. The presented data are mean ± S.E.M. of at least three independent experiments, each performed in triplicate.

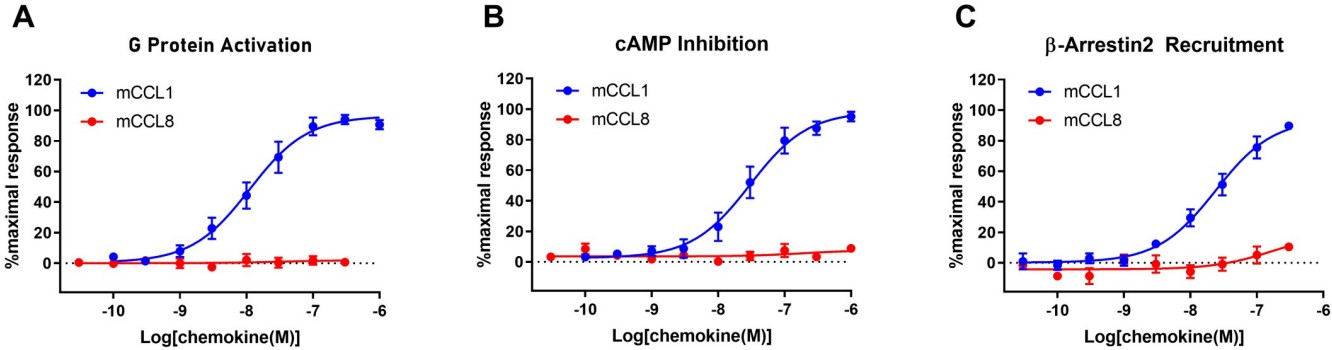

**Fig 7. mCCR8 mediates signal transduction in response to mCCL1 but not mCCL8.** Concentration-response curves for chemokine-induced (A) G protein activation, (B) cAMP inhibition and (C) β-arrestin2 recruitment in Flp-In CHO cells stably expressing mCCR8. The presented data are mean ± S.E.M. of at least three independent experiments, each performed in triplicate.

these CCL18 isoforms (at 300 nM) was able to inhibit the response to 100 nM CCL1 (Fig 6E). These results indicated that the N-terminal variants of CCL18 were neither agonists nor antagonists of CCR8 in this assay system.

## Mouse CCL8 does not stimulate mouse CCR8-mediated signalling in CHO cells

CCL18 is not expressed by mice. However, Islam et al. have reported that murine CCL8 (mCCL8) is an agonist of murine CCR8 (mCCR8) [29] and concluded that mCCL8 is a functional analogue of human CCL18 [23]. Therefore, in light of the above observation that CCL18 does not activate CCR8, we tested the ability of mCCL8 to activate mCCR8 expressed in FlpIn CHO cells. We found that the positive control, murine CCL1 (mCCL1), gave robust, concentration-dependent responses in assays of G protein activation ($EC_{50}$ = 11 nM; $pEC_{50}$ (M) = 7.95 ± 0.12; Fig 7A), cAMP inhibition ($EC_{50}$ = 30 nM; $pEC_{50}$ (M) = 7.52 ± 0.13; Fig 7B) and β-arrestin2 recruitment ($EC_{50}$ = 24 nM; $pEC_{50}$ (M) = 7.62 ± 0.11; Fig 7C). However, mCCL8, up to 300 nM, did not give any significant responses in these assays (Fig 7A–7C).

## Discussion

We undertook this study to evaluate the previous conclusion that CCR8 is a cognate receptor for CCL18 [23]. Commonly, the conclusion that a chemokine and its receptor are a cognate pair is made based on the observation that the chemokine induces robust chemotaxis and/or transduction of G protein-mediated or β-arrestin-mediated signals in a cell line transfected to express the receptor; the transfected cell line should also not endogenously express the receptor in question or, ideally, another high-affinity receptor for the same ligand. The study of Islam et al. [23] had apparently met these criteria by demonstrating that: (a) 4DE4 cells transfected to express CCR8 underwent chemotaxis in response to low nanomolar concentrations of CCL18, whereas untransfected cells did not; (b) the CCL18-stimulated chemotaxis was inhibited by pertussis toxin, indicating it is mediated by Gαi; (c) 4DE4-CCR8 cells underwent calcium flux in response to CCL18, which was desensitized by pre-treatment with the known CCR8 agonist CCL1; (d) CCL18 and CCL1 both induced internalisation of CCR8 on 4DE4-CCR8 cells; and (e) CCL18 and CCL1 competed with each other for binding to 4DE4-CCR8 cells. The conclusion that CCR8 is a cognate receptor of CCL18 was further

supported by experiments in human Th2 cells showing that higher levels of CCR8 (after successive rounds of polarization) correlated with more robust responses to CCL18. Thus, based on the data reported by Islam and co-workers, it was reasonable to conclude that CCR8 is a cognate receptor of CCL18.

The data reported herein clearly contrast with those reported previously. Most directly, using clones of the same 4DE4-CCR8 cell line, cultured under the same (low G418) conditions, we found that low nanomolar concentrations of CCL18 failed to stimulate chemotaxis above background levels, whereas identical concentrations of CCL1 stimulated robust chemotaxis (Fig 2A and 2B). We obtained similar results in both stably- and transiently-transfected L1.2 cells (Fig 2C and 2D). In addition, CCL18 failed to stimulate internalization of CCR8 in the 4DE4-CCR8 cell line, whereas CCL1 stimulated robust CCR8 internalization (Fig 3B). Moreover, FlpIn CHO cells stably transfected with CCR8 gave very robust G protein activation, adenylate cyclase inhibition, and β-arrestin2 recruitment responses to CCL1 but no significant response to CCL18 (Fig 5). These results lead us to conclude that CCR8 alone is not sufficient to act as a functional, high-affinity receptor for CCL18.

Our initial chemotaxis experiments, using the 4DE4-CCR8 cell line, cultured under relatively high G418 conditions, showed a chemotaxis response to high concentrations of CCL18 (Fig 1B). It is possible that this response results from binding of CCL18 to CCR8. However, we consider this unlikely because: (a) the parental 4DE4 cell line gave a chemotaxis response, albeit slightly weaker, to similar CCL18 concentrations; (b) the chemotaxis response of the parental 4DE4 cell line to CCL18 was not inhibited by the CCR8 inhibitor MC148; and (c) 4DE4-CCR8 cells adapted from high to low concentrations of G418 retained the chemotaxis response to CCL1 but completely lost the response to CCL18. Instead, we consider it more likely that the chemotaxis response to high concentrations of CCL18 was mediated by an endogenous, low-affinity receptor to CCL18 that remains to be identified.

In an effort to resolve the discrepancies, we closely examined the similarities and differences between our results and those reported previously. Similar to our results using L1.2-CCR8 cells, Islam et al. [23] noted that L1.2 cells stably transfected with CCR8 yielded only weak chemotaxis or calcium flux responses in their hands, whereas these cells responded more strongly to CCL1. This observation had suggested that L1.2 cells lacked some factor(s) required for the CCR8 response to CCL18, which would be surprising considering that L1.2 cells have been used widely to assay activation of chemokine receptors by their cognate chemokines. However, as noted above, our results using 4DE4-CCR8 cells grown under relatively high G418 conditions, suggest the alternative explanation that 4DE4 cells may endogenously express a relatively low affinity receptor for CCL18. Indeed, the CCL18 chemotaxis data of Islam et al., using 4DE4-CCR8 cells, show a maximal response (using ~10 to ~100 nM CCL18) that is only ~20–25% of the maximal response to CCL1. Thus, it is possible that higher concentrations of CCL18 would have yielded a chemotaxis response approaching the maximal response observed for CCL1, as observed here, which would be consistent with activation of a relatively low affinity receptor. Similarly, Islam et al. reported calcium flux data for 4DE4-CCR8 cells up to a maximal CCL18 concentration of only 25 nM, at which point the response had not reached a plateau, so the $EC_{50}$ and apparent affinity could not be determined. On the other hand, they also showed that unlabelled CCL18 competed with radiolabelled CCL1 for binding to 4DE4-CCR8 cells with a sigmoidal displacement curve and low nanomolar midpoint of inhibition, although the highest concentrations of CCL18 only displaced about half of the bound CCL1. Similarly, unlabelled CCL1 displaced about 60% of radiolabelled CCL18 from 4DE4-CCR8 cells but the displacement curve did not exhibit the expected sigmoidal shape. We wonder whether these competitive binding data may reflect non-specific binding, e.g., to glycosaminoglycans (GAGs), and/or binding to an endogenous receptor expressed in 4DE4

cells. Given that CCL18 has a theoretical isoelectric point of 9.2 as calculated by the Compute pI/Mw tool of Expasy [47], this is a distinct likelihood.

Considering the well-known importance of the N-terminal amino acid residues for the activity of chemokines, we further investigated whether the lack of CCL18 activity observed in signalling assays could be attributed to the use of a different CCL18 isoform. However, none of the four sequence variants tested here was able to stimulate G protein activation or inhibit G protein activation in response to CCL1 (Fig 6D and 6E). It remains possible that a further variant, not studied herein, could activate or inhibit CCR8. However, this would not account for the discrepancies between our results and those reported previously.

CCL18 is only found in primates, so it remains unclear whether rodents have a functionally equivalent chemokine. In addition to reporting the activation of CCR8 by CCL18, Islam and co-workers also reported similar functional data for stimulation of mCCR8 on 4DE4 cells by mCCL8, leading them to conclude that mCCL8 and CCL18 are "functional analogues" [23, 29]. However, in three functional assays for which mCCL1 exhibited robust mCCR8-mediated activity, we were unable to observe any signalling in response to mCCL8 (Fig 7). Thus, the current results cast doubt on the previous conclusions that mCCL8 is an agonist of mCCR8 and that mCCL8 is a functional analogue of CCL18.

The current results may reignite debate regarding the biologically relevant receptor(s) of CCL18. If CCR8 is not a functional receptor, it is worthwhile considering other candidates that could account for the reported activities of CCL18 as a T cell chemoattractant. As noted above, both CCR6 and PITPNM3 have been proposed as potential CCL18 receptors [25, 26]. Subsequent results do not support PITPNM3 as a mediator of CCL18-stimulated leukocyte chemotaxis. However, two very recent studies present evidence that CCR6 mediates the CCL18-stimulated migration of mouse and human T cells [48] as well as a pro-fibrotic phenotype of fibroblasts [49]. CCL18 has also been shown to inhibit activation of CCR3 (confirmed in Fig 4), by binding directly to CCR3, and to inhibit activation of CCR1, CCR2, CCR4 and CCR5 by disrupting presentation of GAG-bound chemokines to these receptors [24, 41]. In addition, the receptor GPR30 was reported to mediate CCL18 inhibition of CXCL12-stimulated chemotaxis on pre-B acute lymphocytic leukemia cells [28, 50]. However, these inhibitory functions do not account for the T cell chemoattractant activity of CCL18. Taken together, the most recent evidence suggests that CCR6 is the most likely candidate as the receptor for CCL18 on T cells.

In conclusion, we have found that, contrary to prior reports, CCR8 does not act as a receptor for CCL18 in several mammalian cell lines and a variety of cellular signalling assays. It remains possible that CCR8 participates in CCL18 signalling in combination with other receptors or signalling factors. Future investigations should focus on the identification of such factors or the validation of alternative receptors for CCL18.

## Acknowledgments

We thank Emma Klein for assistance with protein production, David Steer for collection of mass spectrometry data and staff of the Imperial College Core Flow Cytometry facility for assistance with flow cytometry.

## Author Contributions

**Conceptualization:** Khansa Hussain, Herman D. Lim, Barbara K. Kemp-Harper, J. Robert Lane, Meritxell Canals, James E. Pease, Martin J. Stone.

**Data curation:** J. Robert Lane, Meritxell Canals, James E. Pease, Martin J. Stone.

**Formal analysis:** Khansa Hussain, Herman D. Lim.

**Funding acquisition:** Khansa Hussain, Barbara K. Kemp-Harper, J. Robert Lane, Meritxell Canals, James E. Pease, Martin J. Stone.

**Investigation:** Khansa Hussain, Herman D. Lim, Shankar Raj Devkota.

**Methodology:** Khansa Hussain, Herman D. Lim, Shankar Raj Devkota, J. Robert Lane, Meritxell Canals, James E. Pease, Martin J. Stone.

**Project administration:** James E. Pease, Martin J. Stone.

**Resources:** Martin J. Stone.

**Supervision:** J. Robert Lane, Meritxell Canals, James E. Pease, Martin J. Stone.

**Writing – original draft:** Khansa Hussain, Herman D. Lim, James E. Pease, Martin J. Stone.

**Writing – review & editing:** Khansa Hussain, Herman D. Lim, Shankar Raj Devkota, Barbara K. Kemp-Harper, J. Robert Lane, Meritxell Canals, James E. Pease, Martin J. Stone.

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
