## [Decision Letter · Decision Letter 0]

25 Jun 2024

PONE-D-24-21494The Chemokine Receptor CCR8 is Not a High-affinity Receptor for the Human Chemokine CCL18PLOS ONE

Dear Dr. Stone,

Thank you for submitting your manuscript to PLOS ONE. Both reviewers appreciate your manuscript, however, at least one reviewer and myself found a few points which should be clarified. Therefore, we invite you to submit a revised version of the manuscript that addresses the points raised during the review process. Reference 27 should be edited as the manuscript cannot be found Currently three receptors for CCL18 are reported in the literature GPER (DOI: 10.1002/jcp.22284, 10.21873/anticanres.14291), CCR6 (10.3389/fimmu.2024.1327051; doi:10.3390/cells13030238), and PITPNM3 (doi: 10.1016/j.ccr.2011.02.006). Nevertheless. I agree with your doubts on PITPNM3.

Reviewer 1 raised some additional questions which should be answered.

We look forward to receiving your revised manuscript.

Kind regards,

Gernot Zissel, Ph.D.

Academic Editor

PLOS ONE

Journal Requirements:

"This research was funded by a fellowship from the Government of Saudi Arabia and  King Saud University (K.H.), National Health and Medical Research Council (NHMRC) Project Grant APP1140874 (M.J.S., M.C., J.R.L.) and NHMRC Ideas Grant  2012579 (M.J.S.)."

"We thank Emma Klein for assistance with protein production, David Steer for collection of mass spectrometry data and staff of the Imperial College Core Flow Cytometry facility for assistance with flow cytometry. This research was funded by a fellowship from the Government of Saudi Arabia and  King Saud University (K.H.), National Health and Medical Research Council (NHMRC) Project Grant APP1140874 (M.J.S., M.C., J.R.L.) and NHMRC Ideas Grant  2012579 (M.J.S.)."

"This research was funded by a fellowship from the Government of Saudi Arabia and  King Saud University (K.H.), National Health and Medical Research Council (NHMRC) Project Grant APP1140874 (M.J.S., M.C., J.R.L.) and NHMRC Ideas Grant  2012579 (M.J.S.)."

Reviewers' comments:

Reviewer's Responses to Questions

**Comments to the Author**

1. Is the manuscript technically sound, and do the data support the conclusions?

Reviewer #1: Yes

Reviewer #2: Yes

2. Has the statistical analysis been performed appropriately and rigorously? 

Reviewer #1: Yes

Reviewer #2: Yes

3. Have the authors made all data underlying the findings in their manuscript fully available?

Reviewer #1: Yes

Reviewer #2: Yes

4. Is the manuscript presented in an intelligible fashion and written in standard English?

Reviewer #1: Yes

Reviewer #2: Yes

5. Review Comments to the Author

Reviewer #1: “Negative” results are never easy to publish in peer-reviewed journals and this certainly applies to the present study by Hussain and co-workers. Human CCL18 has been reported to be a second, in addition to CCL1, chemokine ligand for human CCR8, based on in vitro data showing chemotactic activity and direct interaction with CCR8. Furthermore, a recent report in Cell Rep. Med. by the same group provides evidence for in vivo activity of CCL18 for T cells in a human skin transplant model. Interestingly, Figure 4i/j in the original report even demonstrated that human CCL18 is a chemokine ligand for murine CCRR8. Now here, the present study using the same cellular systems (and much more) does not corroborate the initial findings. The study was carefully carried out by multiple laboratories yet did not find any evidence for a high-affinity interaction of CCL18 (and N-terminal truncation variants) with CCR8 or an induction of prototypic cellular responses, including chemotaxis and CCR8 signaling events. These results suggest that CCL18 is at best a low-affinity ligand for CCR8. Considering the importance of CCR8 in pathologies, it is essential that the present findings are made public in the hope that it may instigate future discussions leading to the clarification of this conundrum.

The authors are invited to address the following points.

1) Figure 3 shows that CCL18 was unable to trigger CCR8 internalization. CCL1 was used as a positive control. Are you sure that CCL1 led to CCR8 internalization as opposed to inhibition of Ab 433H binding?

2) CCR8 signaling induces strong Ca2+ mobilization responses. Is there any reason why this assay was not included in the study?

3) In Figure 7, the study was extended to include murine CCR8 and the newly discovered ligand murine CCL8. I am not certain how these data help to resolve the human CCL18 questions. Perhaps it would be best to more carefully examine the murine chemokine system in a separate study?

4) Overall, the “negative” data are convincing, and it is puzzling that the results obtained with CCR8-transfected 4DE4 cell lines by the two groups gave such conflicting results. The striking discrepancies beg for a scientific explanation?

Reviewer #2: In their manuscript Hussain et al describe a detailed evaluation of the potential (previously reported) interaction between human CCL18 and the chemokine receptor CCR8 as well as the potential interaction of this receptor with murine CCL8. CCL18 is a highly abundant chemokine in blood and upregulated in multiple pathologies. As such, it is crucial that we understand its interaction with potential chemokine receptors. Authors convincingly show using multiple cell lines that human CCL18 and murine CCL8 do not signal through the human CCR8. The experiments have been carefully performed, confirmed in multiple assays and on multiple cell lines using a broad concentration range of the potential ligand. In addition, since CCL18 might be processed by proteases, authors also investigated whether proteolytic processing might activate CCL18 on CCR8. However, also the processed CCL18 forms (as well as murine CCL8) failed to signal through CCR8.

6. PLOS authors have the option to publish the peer review history of their article (what does this mean?). If published, this will include your full peer review and any attached files.

Reviewer #1: **Yes: **Bernhard Moser

Reviewer #2: No

---

## [Author Response · Author response to Decision Letter 0]

8 Aug 2024

See attached Response to Reviewers file.

---

## [Decision Letter · Decision Letter 1]

29 Aug 2024

The chemokine receptor CCR8 is not a high-affinity receptor for the human chemokine CCL18

PONE-D-24-21494R1

Dear Dr. Stone,

We’re pleased to inform you that your manuscript has been judged scientifically suitable for publication and will be formally accepted for publication once it meets all outstanding technical requirements.

Kind regards,

Gernot Zissel, Ph.D.

Academic Editor

PLOS ONE

Additional Editor Comments (optional):

Reviewers' comments:

Reviewer's Responses to Questions

**Comments to the Author**

1. If the authors have adequately addressed your comments raised in a previous round of review and you feel that this manuscript is now acceptable for publication, you may indicate that here to bypass the “Comments to the Author” section, enter your conflict of interest statement in the “Confidential to Editor” section, and submit your "Accept" recommendation.

Reviewer #1: All comments have been addressed

2. Is the manuscript technically sound, and do the data support the conclusions?

Reviewer #1: Yes

3. Has the statistical analysis been performed appropriately and rigorously? 

Reviewer #1: Yes

4. Have the authors made all data underlying the findings in their manuscript fully available?

Reviewer #1: Yes

5. Is the manuscript presented in an intelligible fashion and written in standard English?

Reviewer #1: Yes

6. Review Comments to the Author

Reviewer #1: Thanks for the revisions. Thanks for the revisions. Thanks for the revisions. Thanks for the revisions.

7. PLOS authors have the option to publish the peer review history of their article (what does this mean?). If published, this will include your full peer review and any attached files.

Reviewer #1: **Yes: **Bernhard Moser

---

## [Editor Report · Acceptance letter]

2 Sep 2024

PONE-D-24-21494R1 

PLOS ONE

Dear Dr. Stone, 

I'm pleased to inform you that your manuscript has been deemed suitable for publication in PLOS ONE. Congratulations! Your manuscript is now being handed over to our production team.

Kind regards, 

on behalf of

Prof. Dr. Gernot Zissel 

Academic Editor

PLOS ONE